# Complementary Therapeutic Effect of Fecal Microbiota Transplantation in Ulcerative Colitis after the Response to Anti-Tumor Necrosis Factor Alpha Agent Was Lost: A Case Report

**DOI:** 10.3390/biomedicines12040800

**Published:** 2024-04-03

**Authors:** Jongbeom Shin, Ga Hyeon Baek, Boram Cha, Soo-Hyun Park, Jung-Hwan Lee, Jun-Seob Kim, Kye Sook Kwon

**Affiliations:** 1Department of Internal Medicine, Inha University College of Medicine, Incheon 22332, Republic of Korea; shinjongv@inha.ac.kr (J.S.); chaboram@inha.ac.kr (B.C.); 2Department of Nano-Bioengineering, Incheon National University, Incheon 22012, Republic of Korea; inu202221153@inu.ac.kr; 3Department of Neurology, Soon Chun Hyang University Hospital Seoul, Seoul 04401, Republic of Korea; g2skhome@gmail.com; 4Division of Gastroenterology, Department of Hospital Medicine, Inha University Hospital, Incheon 22332, Republic of Korea

**Keywords:** ulcerative colitis, anti-TNFα, loss of response, gut microbiome, FMT

## Abstract

In patients with ulcerative colitis (UC), the development of an antidrug antibody (ADA) to anti-tumor necrosis factor (TNF)α agent is a crucial problem which aggravates the clinical course of the disease, being cited as one of the most common causes for discontinuing anti-TNFα treatment. This is due to ADA eventually causing secondary LOR, leading to discontinuation of anti-TNFα treatment. Recently, research on the microbiome and relationship between worsening UC and dysbiosis has been conducted. Further, investigations on the association between the microbiome and secondary LOR are increasing. Here, we present the therapeutic effect of fecal microbiota transplantation (FMT) on a 42-year-old man with secondary LOR and high ADA levels. FMT has recently been used for the treatment of, and for overcoming, drug resistance through microbiome modification. Stool samples were collected from the patient before and 4 weeks after FMT. Symptoms, including hematochezia and Mayo endoscopy sub-scores, improved after FMT, while ADA levels decreased by one-third to less than half the value (29 ng/mL) compared to before FMT (79 ng/mL). Additionally, the trough level of infliximab became measurable, which reflects the improvement in the area under the concentration (AUC). *Butyricicoccus*, *Faecalibacterium*, *Bifidobacterium*, *Ligilactobacillus*, *Alistipes*, and *Odoribacter*, which regulate immune responses and alleviate inflammation, also increased after FMT. We report a case in which microbiome modification by FMT increased the AUC of anti-TNFα in a patient who developed secondary LOR during anti-TNFα treatment, thereby improving symptoms and mucosal inflammation.

## 1. Introduction

Ulcerative colitis (UC) is a chronic disease with relapsing−remitting symptoms, such as hematochezia or mucoid stool, urgency, abdominal discomfort, and other extraintestinal manifestations [1,2,3,4]. It is known that genetic factors, host immune system, intestinal microbiota dysbiosis, and environmental factors contribute to the development of ulcerative colitis [5]. Clinical response and clinical remission are early targets of UC treatment, while endoscopic healing is a long-term target [6].

In patients with moderate-to-severe UC that do not respond to immunosuppressants, biologics are administered to achieve clinical remission and mucosal healing. The occurrence and severity of ulcerative colitis are positively correlated with the overpopulation of *Eubacterium rectum*, *E. coli*, and *Ruminococcus gnavus*, which cause cellular inflammation. Recently, various biologics and small molecules that can be used to treat ulcerative colitis have been developed and administered [7]. Among them, the biological agent that was developed early and is still widely used is infliximab (IFX), an anti-TNFα agent [8].

The an anti-drug antibody (ADA) may be arise during necessary for treatment of the anti-tumor necrosis factor (TNF)α agent, which is a major problem that worsens the clinical course of patients with UC. It is a leading cause for treatment discontinuation due to secondary loss of response (LOR) to IFX treatment [8,9,10]. Since secondary LOR cannot be restricted after it occurs, measures to prevent its occurrence are being investigated. The most widely used treatment method to suppress ADA is the concomitant administration of azathioprine with an anti-TNFα agent [11]. However, because overcoming LOR with azathioprine is insufficient and often fails, interest in the relationship between the microbiome and secondary LOR has increased.

Fecal microbiota transplantation (FMT) is widely accepted as the most effective treatment for *Clostridium difficile* infection and has been extensively studied for the treatment of pathogenic bacterial infections, chronic inflammatory diseases, immune regulation, management of metabolic disorders and obesity, as well as immune system support [12,13,14,15]. It is also effective at suppressing inflammation in patients with inflammatory bowel disease, including those with UC [16]. FMT also affects immunological responses, which are associated with a substantial reduction in colonic mucosal CD8+ T cell density and reduced serum concentrations of IL-6 and IP-10. In particular, serum levels of IL-6 and VCAM-1 were significantly correlated with CRP and ESR [17]. This immune-modulating ability of FMT leads to the positive effects of the treatment in the treatment of ulcerative colitis. Until recently, ten randomized controlled trials of FMT have been conducted for UC, and over 400 patients have been administered FMT [18,19,20,21,22,23,24]. Haixia Liu et al. reported a significant advantage in inducing clinical and endoscopic remission using FMT compared to a placebo in a re-meta-analysis (OR 3.83 [2.31, 6.34]) [20]. Another report showed that the drug response improved after FMT even in patients with melanoma and colorectal cancer who were resistant to anti-PD-1 treatment [25,26,27,28,29,30]. Yuri Gorelik et al. observed that ADA production differed significantly depending on the type of antibiotic administered, reporting that ADA production was related to the microbial composition [30]. Therefore, research is actively underway to elucidate the effects and microbiome modulation mechanisms of FMT on changes in drug responses and immune responses [26,27,28,29].

We report a case in which reduction in ADA to anti-TNFα was achieved through FMT in a UC patient with secondary LOR (who could not use azathioprine due to previous neutropenic fever), achieving clinical remission and improvement of endoscopic activity.

## 2. Case Report

A 42-year-old man was diagnosed with ulcerative pancolitis in June 2018. After clinical remission was induced, azathioprine (1 mg/kg) was started to taper off the steroid. However, the drug had to be discontinued after 3 months because of a diagnosis of leukopenia accompanied by neutropenic fever. Eventually, the patient was administered infliximab (an anti-TNFα agent) at induction doses and went into clinical and endoscopic remission. Remission with infliximab was maintained for more than 3 years. However, the patient complained of abdominal pain, diarrhea, and hematochezia, which worsened 4–6 weeks after infliximab administration. In October 2021, the patient presented relapse, which was confirmed using endoscopy (two points on the Mayo endoscopic subscore). As a result of checking the calprotectin to establish a level that reflects the endoscopic findings, a high elevation was confirmed. The results of a stool analysis performed to check for evidence of bacterial or viral infection were negative (including *C. difficile*).

In order to find the pharmacokinetic cause of the worsening of patient’s symptoms and endoscopic findings, the infliximab trough level was measured. After ADA developed (79 ng/mL) and the trough level was not detected (<0.1 μg/mL), a secondary LOR was eventually diagnosed. The need to change infliximab to other biologics was explained to the patient, but the patient wished to continue infliximab because clinical remission was maintained for up to 6 weeks after the scheduled administration. Although azathioprine is the most effective treatment for controlling LOR, because the patient had previously developed adverse reactions to azathioprine, that treatment method could not be used. The patient was recommended another anti-TNFα drug (adalimumab), which they refused because of fear of subcutaneous injection. A change in medication to a different type of biological agent was recommended, but the patient also refused this because some improvement was obtained after infliximab treatment. Therefore, we had an in-depth discussion with the patient regarding additional inflammation control methods and decided to perform an FMT.

The FMT donor was a 39-year-old man without any gastrointestinal or other health problems. All donors underwent blood chemistry, endoscopic, and stool tests, being selected if they met the standards of the FMT guidelines. From the donor, 60 g of donated stool was filtered to prepare a fecal suspension and diluted in a glycerol−saline solution (12.5% glycerol in 0.90% *w*/*v* NaCl in water). After processing, the fecal suspension was stored at −80 °C. For performing the FMT procedure, stored fecal suspension was thawed for 4 h prior to FMT at room temperature. The FMT was administered via colonoscopy after bowel preparation.

The FMT was performed 4 weeks after infliximab administration, considering the point of decrease in the infliximab effect. After the 1st FMT, the patient’s clinical remission was maintained until the next dose of infliximab without an increase in the number of diarrhea or bloody stools observed 4 weeks after infliximab administration. To intensify the anti-inflammatory effect and modify the patient’s microbiome, a 2nd FMT was performed after 8 weeks from the 1st FMT. At the time of the 2nd FMT, the Mayo endoscopic subscore was 1 and all erosive lesions had improved. Further, the ADA level was reduced by less than half compared to before FMT (79 ng/mL to 29 ng/mL) and the infliximab trough level was confirmed to be 0.1 μg/mL. As clinical remission and endoscopic improvement were observed (Figure 1), an additional FMT was not performed, and remission was maintained for 6 months; however, abdominal pain and bloody stools recurred. The ADA level increased by 100 ng/mL compared to the ADA level after FMT (from 29 ng/mL to 129 ng/mL) and ulcerative lesions were observed upon endoscopic evaluation. Based on the findings, inflammation was estimated to be difficult to control with only additional FMT treatments without a switch of biologic agent, so it was decided to change the dosage to vedolizumab (a recombinant humanized immunoglobulin G1 monoclonal antibody directed against the human lymphocyte α4β7 integrin). The patient is now currently in remission (December 2023).

Fecal samples were collected for microbiome analysis from the donor and UC patient (3 days before the first FMT; pre-FMT, 4 weeks after FMT; post-FMT). All fecal samples were stored at −80 °C immediately after collection until the fecal deoxyribonucleic acid extraction was done. The V3-V4 sections of the 16s rRNA gene were amplified using the following primers: 341F-TCGTCGGCAGCGTCAGATGTGTATAAGAGACAGCCTACGGGNGGCWGCAG and 805R-GTCTCGTGGGCTCGGAGATGTGTATAAGAGACAGGAC TACHVGGGTATC TAATCC. The amplicons were sequenced on a MiSeq platform (Illumina, San Diego, CA, USA), and the sequence data were processed using Mothur v.1.47.0. (http://www.mothur.org/) and the SILVA reference database following the Mothur Miseq SOP [31].

The relative abundances of *Actinobacteriota*, *Firmicutes*, *Bacteroidota*, and *Proteobacteria* at the phylum level recovered from the patient were of levels similar to that of the donor (Figure 2A) after FMT. At the family level, *Bacteroidaceae* and *Butyricicoccaceae*, which were not observed in patients with UC before FMT, also showed significant recovery (Figure 2B). Furthermore, the microbial alpha diversity assessed using the Shannon index appeared to increase after FMT (Figure 2C). Regarding beta diversity, the microbiota membership became more similar to that of the donor after the FMT, with a greater resemblance being observed after the 2nd FMT (Figure 2D).

## 3. Discussion

Treatment or control of ADA when patients with secondary LOR to infliximab are not responsive to azathioprine has not yet been identified. This clinical case is a notable case report which describes the improvement of clinical features and decreased ADA to anti-TNFα in a patient with secondary LOR after FMT. In addition, our results showed that undetectable low levels of anti-TNFα increased to a measurable range after FMT. This result means that the area under concentration (AUC) of infliximab was increased after FMT. The results also showed that the microbial diversity in the patients improved and was similar to that of the donor after FMT, confirming the assumption that the effect was due to microbiome modification.

Treatment with FMT in UC is still controversial. AGA guidelines published in 2024 suggested that conventional FMT should not be used in adults with UC but could reasonably be used in the context of clinical trials or potentially outside of clinical trials when no comparable or satisfactory alternative treatment is available [32]. The following reasons were provided to deter recommendations: Significant heterogeneity in FMT administration was observed in the studies performed, as the study sizes were insufficient and most FMTs were administered in concomitant therapy; however, studies showing the effectiveness of FMT continue to be reported [16,23,24,33]. In the largest cohort study, including 259 UC patients that confirmed the effectiveness of washed microbiota transplantation, a clinical response rate of 70.7% was achieved [33]. In this cohort study, 29.7% of patients achieved steroid free clinical remission after 6 months. Additionally, the effectiveness of FMT compared to the control group for inducing remission of ulcerative colitis has been reported to range from 25% [23] to 45% [24]. In summary, the effectiveness of FMT in UC patients cannot be denied, although controversy continues due to there still being insufficient evidence to establish it as a treatment. Therefore, FMT must be considered selectively when choosing it as a treatment method in UC patients. In this report, there was no comparable or satisfactory alternative treatment after the LOR of infliximab was received, and FMT was performed for complementary therapy purposes simultaneously with infliximab, a proven UC treatment.

Several risk factors affect the production of ADA [34,35,36]. Regular drug administration and concomitant immunosuppressive agent administration are recommended to suppress ADA. However, a recent retrospective multicenter study reporting LOR rates in IBD patients found the LOR rates per patient year in UC to be as high as 9% [11,37]. In Crohn’s Disease, intensification of anti-TNF alpha agent doses or the addition of immunomodulators may be considered to overcome LOR [38]. On the other hand, in patients with UC undergoing anti-TNFα treatment and who have ADA, there are limited options for treatment beyond altering the medication. Due to secondary LOR, approximately 50% of UC patients with anti-TNFα treatment experience worsening inflammation, eventually leading to a change in medications [39]. Therefore, the finding that microbiome modification through FMT can increase the AUC of infliximab by decreasing ADA and measurably altering anti-TNFα is probably a clinically meaningful change.

We characterized the microbiome to explore the underlying reason for a decrease in ADA levels after FMT. The LEfSe analysis revealed that a relative abundance of *Butyricicoccus*, *Faecalibacterium*, *Bifidobacterium*, *Ligilactobacillus*, *Alistipes*, and *Odoribacter* genera increased with FMT. *Butyricicoccaceae* have mucosal anti-inflammatory functions and are associated with short-chain fatty acid (SCFA) production [40]. *Faecalibacterium* are known to prevent or improve clinical symptoms related to UC by inhibiting the loss of CD4+ CD25+ Regulatory T cells (Tregs) in the spleen, thereby regulating the immune response [41]. Treg cells also play a crucial role in suppressing autoimmunity and maintaining immune tolerance [42]. *Bifidobacterium* and *Ligilactobacillus* are well-known probiotic bacteria [43,44]. Conversely, the genera that decreased after FMT were *Blautia*, *Streptococcus*, *Actinomyces*, and *Pseudomonas*. A previous study has shown that a reduction in *Blautia* species is associated with an improvement of patients with UC [45]. An increase in the proportion of *Streptococcus* may be a significant factor in the onset of UC [46]. As found in other studies, the abundance of *Actinomyces* and *Pseudomonas* decreased following FMT treatment [47,48] (Figure 2E). Taken together, the gut microbiota analysis suggests that FMT significantly influences the gut microbiota, and these changes might be triggered by immunological effects.

In the present case report, after the second FMT, the characteristics of the microbiome were more similar to that of the donor (compared to after the first FMT). This finding provides evidence that repeated FMT treatment enhances the improved modification of the recipient microbiome. In a study targeting IBS patients, Lee et al. concluded that repeating FMT would be effective, as long-term colonization of beneficial bacteria is positively associated with symptom improvement [49]; therefore, we recommended the patient undergo an additional third FMT. However, the patient refused this treatment because it was difficult to visit the hospital every month to maintain FMT and infliximab administration at 4-week intervals, in addition to it being too painful to repeat the bowel preparation process for FMT. Due to patient refusal, we could not determine whether multiple FMTs further improved the abundance of the beneficial microbiome. Additional research is required to determine whether regular microbiome testing using an FMT suppresses LOR in patients with ADA.

This case report has limitations. First, it was assumed that the trough level of infliximab increased and the AUC increased; however, it was not possible to indicate how much the AUC had improved. If serial infliximab serum levels had been measured, clear values could have been provided, but the serum of the case patient could not be obtained serially. It is necessary to conduct studies to determine the effect of FMT on the AUC of infliximab in UC patients with LOR. Second, the results of changes in microbiome composition and ADA levels up to the point when the patient’s inflammation worsened 6 months after performing two FMTs are unknown. Additional research will be needed to determine which changes in microbiome composition affect the clinical aspects of patients and the level of ADA.

## 4. Conclusions

Although the quality of life of UC patients have improved with the administration of an anti- TNFα agent, a large number of patients suffer from LOR due to the development of ADA. Until recently, methods for overcoming LOR by reducing ADA levels have been ineffective. Under these circumstances, FMT decreased ADA levels and increased AUC, ultimately improving clinical symptoms and endoscopic findings. This case report suggests that there may be an association between the gut microbiota and secondary LOR of IFX in patients with UC. Furthermore, we provided evidence on the potential of FMT as a complementary treatment option through an increase in AUC for patients with UC who develop LOR. Further studies on the therapeutic effects of FMT and the microbiome characteristics in patients with ADA-related LOR are required.

## Figures and Tables

**Figure 1 biomedicines-12-00800-f001:**
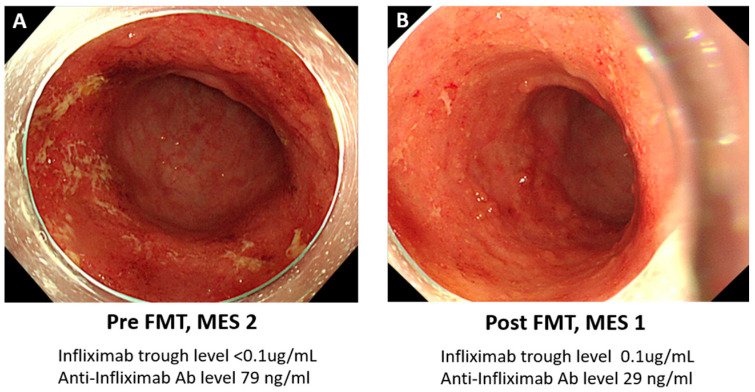
**Comparison of Mayo Endoscopic Subscore (MES) before FMT and after FMT** (**A**) Pre-FMT, Mayo endoscopic subscore was 2. (**B**) Post-FMT, Mayo endoscopic subscore was 1.

**Figure 2 biomedicines-12-00800-f002:**
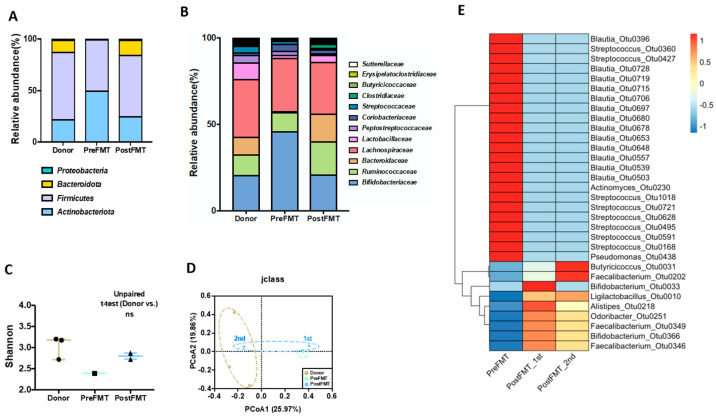
**Comparison of intestinal microflora of donor, pre-FMT, post-FMT groups.** (**A**) Relative abundance bar chart of phylum level. Only phyla levels corresponding to 1% or more were represented. (**B**) Relative abundance bar chart of family level. Only family levels corresponding to 1% or more are represented. (**C**) Alpha diversity comparisons of microbial communities of Donor, Pre-FMT, Post-FMT group. Unpaired *t*-tests (two-tailed) were used to analyze variations between the two groups. (**D**) Beta diversity comparisons of microbial communities of Donor, Pre-FMT, Post-FMT group. Principal Coordinate Analysis (PCoA) plots of jclass and thetayc matrixes were used to assess community membership and structure similarity. The axis labels indicate the percentage occupied by each principal coordinate. (**E**) Heatmap of OTUs that exhibited significant differences Pre-FMT and Post-FMT in LEfSe analysis. Red and blue indicate a higher and lower abundance, respectively.

## Data Availability

Data supporting the findings of this study are available from the first author [J.S.] upon reasonable request.

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
