# Peer review of "Complementary Therapeutic Effect of Fecal Microbiota Transplantation in Ulcerative Colitis after the Response to Anti-Tumor Necrosis Factor Alpha Agent Was Lost: A Case Report"

_biomedicines, 2024, doi:10.3390/biomedicines12040800_

Round 1
Reviewer 1 Report
Comments and Suggestions for Authors
The case report "Complementary Therapeutic Effect of Fecal Microbiota Transplantation in Patients with Ulcerative Colitis Who Developed a Secondary Loss of Response to Anti-Tumor Necrosis Factor Alpha: A Case Report," is of particular importance due to its niche subject matter. However, I would recommend a few revisions:
1. The introduction should provide a more comprehensive overview, emphasizing multiple studies related to microbiota transfer in inflammatory bowel disease. To enhance this aspect, you can directly reference review studies that have systematically compiled significant works from the literature. For example: https://doi.org/10.3390/biomedicines11041016
2. It is not clear at what point you collected the stool for microbiota analysis.
3. What were the factors that could have interacted with the results of microbiota transfer?
4. In the discussion chapter, add more literature studies.
5. It would be advisable to have the conclusions chapter separate.
Comments on the Quality of English LanguageMinor editing of English language required
Author Response
I would like to extend my sincere appreciation for your diligent review and insightful comments on my manuscript. Your expertise and thoughtful suggestions have significantly enhanced the quality of my work. Your constructive feedback has been invaluable in refining the content, clarifying ideas, and strengthening the overall presentation. I am genuinely grateful for your time and effort invested in helping me improve this research.
For the details, Please see the attachment.

Reviewer 2 Report
Comments and Suggestions for Authors
I appreciate this case report and the work the Authors’ team invested in preparing and writing the manuscript. Unfortunately, reading and properly understanding the paper are hampered by the poor quality of the English language. I suggest the Authors to contact a professional translation service, also with good knowledge of the medical language. Normally, I would not start with commenting on the language; however, in this case, it is a pity and does not favor the manuscript at all. There are too many mistakes to mention them point-by-point: use of plural instead of singular, long sentences and too wordy, repetitions, missing adverbs, conjunctions, subject-verb disagreement etc.
I have listed some comments/suggestions for consideration below:
1. Title – should be reformulated in order to be better understood.
Alternative: “Complementary Therapeutic Effect of Fecal Microbiota Transplantation in Ulcerative Colitis, after the response to Anti-Tumor Necrosis Factor Alpha Agent was lost: A Case Report” or something like that. Just to be clear that the loss of response appeared before the FMT.
After reading JUST the Abstract, I was thinking of: “Dual beneficial effect of Fecal Microbiota Transplantation in Ulcerative Colitis – improvement of disease activity and recovery of the loss of response to Anti-Tumor Necrosis Factor Alpha: A Case Report”.
However, after reading the full manuscript, I do not think this would be correct. The level of IFX was still very low – from undetectable to 0.1 microg/mL. Improvement in disease activity appeared probably because of the FMT and the patient kept remission for 6 months. I understand that the level of anti-IFX diminished, but this did not result in normalization of the trough level of IFX.
Even the Authors wrote in the manuscript : “The results also showed that the microbial diversity in the patients improved and was similar to that of the donor after FMT, confirming the assumption that the effect was due to microbiome modification”.
Please revise.
2. Abstract
a. Please correct: “It is the most common reason for discontinuing anti-TNFα treatment because of secondary loss of response (LOR).”
b. Please delete “Since secondary LOR cannot improve ADA after it occurs, only clinical manifestations are considered to prevent the development of ADA.”, as it does not make any sense.
c. “Further investigations on the association between the microbiome and secondary LOR are increasing”. Please delete from here and correct. And, in the main text, please elaborate on this.
d. Please place this sentence before the case presentation. “FMT has recently been used for the treatment of and for overcoming drug resistance through microbiome modification.”
e. Please present not only that “Mayo endoscopic sub-score improved”, mention also about symptoms. In the main text, it is written “clinical remission”.
f. “In comparison with before FMT (79 ng/mL), the ADA levels decreased to one-third level (29 ng/mL); well – 29 x 3 = 87; please correct. Moreover, in the main text, the Authors wrote : “ADA level was reduced by less than half”.
g. “and the infliximab trough level increased” – 0.1 microg/mL represents a minimal increase, still incredibly low. We need a good trough level of IFX (greater than 5 and even more, according to recent data), in order for this to be able and act on the disease.
h. Conclusion: Please revise and rephrase conclusion “Our findings....”
3. Introduction
a. Please replace reference [1] with a recent one. A plethora does exist.
b. “A long-term clinical remission is the primary goal of treating UC”. This is wrong – Please read and insert STRIDE-II reference (Turner D, et al 2021). It is endoscopic healing!
c. “The most effective and widely used biologic agent used for treatment of UC is infliximab (IFX) which is an anti-TNFα agent.”. Please revise and insert reference, as other biologics have been found to be more effective, very recently.
c. Please replace references [3-5] with recent ones and correct ones. Ref. [3] is about Crohn’s disease, not UC. Ref. [4] does not refer to IBD.
d. “Until recently, ten randomized controlled trials of FMT have been conducted for UC, and over 400 patients have been administered FMT [12-14]. Please update!
(e.g.
* Liu X, Li Y, Wu K, Shi Y, Chen M. Fecal Microbiota Transplantation as Therapy for Treatment of Active Ulcerative Colitis: A Systematic Review and Meta-Analysis. Gastroenterol Res Pract. 2021 Apr 23;2021:6612970.
* Nabil El Hage Chehade, Sara Ghoneim, Sagar Shah, Anastasia Chahine, Fadi H Mourad, Fadi F Francis, David G Binion, Francis A Farraye, Jana G Hashash, Efficacy of Fecal Microbiota transplantation in the Treatment of Active Ulcerative Colitis: A Systematic Review and Meta-Analysis of Double-Blind Randomized Controlled Trials, Inflammatory Bowel Diseases, Volume 29, Issue 5, May 2023, Pages 808–817.
* Wei ZJ, Dong HB, Ren YT, Jiang B. Efficacy and safety of fecal microbiota transplantation for the induction of remission in active ulcerative colitis: a systematic review and meta-analysis of randomized controlled trials. Ann Transl Med. 2022 Jul;10(14):802.
* Liu, H., Li, J., Yuan, J. et al. Fecal microbiota transplantation as a therapy for treating ulcerative colitis: an overview of systematic reviews. BMC Microbiol 23, 371 (2023).)
e. Line 59 – I said there were too many mistakes, in order to be mentioned point-by-point, but please correct “systemic review”.
f. Please elaborate with references - studies regarding gut microbiota and effects on the response to Anti-TNF. The Authors wrote :“Therefore, research is actively underway to elucidate the effects and microbiome modulation mechanisms of FMT on changes in drug responses and immune responses”.
g. Aim: “We report a case study in which improvement of clinical symptoms” (in the main text, it is mentioned clinical remission”; please insert also “improvement of endoscopic activity”.
4. Case Report
a. How was the remission induced? For sure, not with AZA, it is not an agent for induction of remission.
b. “However, the patient complained of abdominal pain, diarrhea, and hematochezia, which worsened 4–6 weeks after infliximab administration. In October 2021, the patient presented relapse”; well, the relapse was presented in the previous sentence. Please revise and, since you mentioned October 2021, please also write clearly the month of the diagnosis (in 2018).
c. “Calprotectin levels were elevated” – no relevance, since the patient had hematochezia and endoscopic activity. Sure, it must have been elevated, but it was done for nothing. Just waste of money.
d. “ Donors underwent blood chemistry, endoscopic, and stool tests and were selected if they met the standards of the FMT guidelines” Were there more donors and the presented one was chosen?
e. Please clarify: “FMT was performed in week four of infliximab administration” (is it after?), “considering the point of decrease of the infliximab effect, and after the 1st FMT.”. How come after the first FMT?
f. “> 29 microg/mL “ means more than 29?
g. “Infliximab trough level was confirmed to be 0.1 ug/mL”. As mentioned before, it is still very very low.
h. “ADA level increased more than three times after FMT (129 ng/mL) “ – 29 x 3 = 87; please clarify.
i. “Fecal samples were collected for microbiome analysis from the donor and UC patient (pre-FMT and post-FMT)”. When post FMT? It seems it was assessed twice (from the figure as well) – but when?
k. Please mention statistics for Fig. 2A and Fig. 2B
l. Fig. 2 A – I see only Actinobacteriota, Firmicutes, Bacteroidota.
m. Fig. 1B – I do not see all colors representing the family level.
n. Fig. 2E must be introduced in the “Case report”. Not in “DIscussion”, as it refers to the case and not studies from the literature.
5. Discussion
a. Please delete: “Treatment or control of ADA when patients with secondary LOR to infliximab are not responsive to azathioprine has not yet been identified.”
b. “This clinical case is a notable and interesting case report”– please keep only “notable”.
c. “In addition, our results showed that undetectable low levels of anti-TNFα increased to a measurable range after FMT." But still very low! It would have been interesting to measure the levels further (during the 6 months of clinical remission).
d. Please remove “active inflammation control”, as it does not make any sense (lines 153-154)
e. “However, currently no suitable method is available for reducing prevalent ADA in a patient...”. please avoid redundancy.
f. “Therefore, a decrease in ADA and an increase in the level of anti- TNF-alpha along with clinical relief after microbiome modification through FMT has great clinical significance”. Please clarify and explain. What increase in the level of anti- TNF?
g. “Another study showed a decrease in the abundance of Actinomyces and Pseudomonas after FMT treatment [28, 29] (Figure 2E). Please correct. Fig. 2E refers to the case. Moreover, there are two studies, not one. Maybe you wanted to write that your results were found also in other studies...
h. In “Case Report”, it was written : “inflammation was estimated to be difficult to control with only additional FMT without a switch of biologic agent...”. However, in Discussion, it was mentioned that “However, the patient who improved after two FMTs refused to undergo procedures such as bowel preparation to receive additional FMT...”. Which one is correct, please? Very confusing.
i. Conclusion: “although the quality of life of UC patients have improved with the administration of anti- TNFα agent, a large number of patients suffer from LOR.” This is not the conclusion of this case report.
k. “Until recently, methods for overcoming LOR by reducing ADA levels have been ineffective.”. Not really true for every case (e.g. intensification, adding an immunomodulator).
l. “Furthermore, we provided evidence for the potential of FMT as a complementary treatment option for patients with UC who develop LOR.” I am not very convinced.
6. References: Except for the references regarding microbiota (and just in “Discussion”), most of them are old. Please update.
Comments on the Quality of English LanguageThe quality of the English language is poor.
Author Response

(The authors gave the same response as above.)

Reviewer 3 Report
Comments and Suggestions for Authors
Comments to the Authors of manuscript number: biomedicines-2865873 entitled “Complementary therapeutic effect of fecal microbiota transplantation in patients with ulcerative colitis who developed a secondary loss of response to anti-tumor necrosis factor alpha: A case report”.
The paper presents the problem of ulcerative colitis (UC) and its treatment, particularly focusing on the issue of secondary loss of response (LOR) to infliximab (IFX) due to the development of anti-drug antibodies (ADA). It discusses the challenges associated with managing UC patients who do not respond to immunosuppressants and introduces fecal microbiota transplantation (FMT) as a potential treatment option. The case report describes the treatment course of a 42-year-old male diagnosed with ulcerative pancolitis who initially responded well to infliximab but later developed secondary loss of response (LOR) due to anti-drug antibodies (ADA). The patient's history, including previous treatment with azathioprine and subsequent adverse reactions, is detailed, providing context for the decision-making process. The decision to proceed with fecal microbiota transplantation (FMT) as an alternative treatment after the patient's reluctance to switch to other biologic agents is explained. The donor selection process and FMT procedure are outlined, demonstrating adherence to established guidelines for FMT.
1. The report documents the timing of FMT administration relative to infliximab dosing and describes the clinical and endoscopic improvements observed after FMT, as well as the subsequent recurrence of symptoms and need for a change in treatment to vedolizumab.
2. Microbiome analysis before and after FMT is conducted, revealing changes in microbial composition and diversity following FMT. The results suggest a shift towards a microbiota profile more closely resembling that of the donor post-FMT.
3. Overall, the case report provides a comprehensive account of the patient's treatment journey, including rationale for treatment decisions, procedural details, and clinical outcomes. However, further discussion on the potential factors contributing to the recurrence of symptoms after FMT and the decision-making process leading to the switch in treatment to vedolizumab could enhance the depth of the analysis. Additionally, addressing any limitations or challenges encountered during the treatment process would provide a more balanced perspective.
4. The discussion effectively summarizes the key findings of the case report, including the improvement of clinical symptoms, decreased ADA levels, and changes in the microbiome composition observed after FMT.
5. It provides context by discussing the limitations of current treatment options for UC patients with secondary LOR to infliximab and the need for alternative therapeutic approaches.
6. The discussion elaborates on the potential mechanisms underlying the observed changes in ADA levels and microbiome composition after FMT, offering insights into the immunological effects of microbiome modification.
7. The discussion could benefit from a more detailed analysis of the specific mechanisms by which FMT may lead to decreased ADA levels and improved treatment outcomes in UC patients. Providing more in-depth insights into the interactions between the gut microbiota, immune system, and drug response could enhance the comprehensiveness of the discussion.
8. While the discussion highlights the changes in microbial composition observed after FMT, it would be valuable to discuss the potential clinical implications of these changes in relation to UC pathogenesis and treatment response. Exploring how specific microbial taxa may influence disease activity and therapeutic outcomes could provide additional context for interpreting the findings.
9. The conclusion could be strengthened by summarizing the implications of the study findings for clinical practice and future research directions more explicitly. Clearly outlining the potential role of FMT as a complementary treatment option and the need for further studies to validate its efficacy and elucidate underlying mechanisms would enhance the overall impact of the discussion.
Author Response

(The authors gave the same response as above.)

Reviewer 4 Report
Comments and Suggestions for Authors
This present case report clearly emphasized the specific effectiveness of FMT. The use of FMT in the treatment of UC is obvious controversy. The largest UC cohort to date (n=259) showed effectiveness (Older patients benefit more from sequential courses of washed microbiota transplantation than younger population with ulcerative colitis. Scand J Gastroenterol. 2023;58(8):890-899), but the latest expert opinions (AGA Clinical Practice Guideline on Fecal Microbiota-Based Therapies for Select Gastrointestinal Diseases. Gastroenterology. 2024 Mar;166(3):409-434) did not recommend this. So, it is necessary for the authors to discuss this in order to provide readers with a deeper understanding.
Author Response
Thanks for you comments. Please see the attachment.

Round 2
Reviewer 1 Report
Comments and Suggestions for Authors
The revised version of the case report was not written properly.
- The introduction is indeed improved.
- The case is presented in an inappropriate manner. Events are mixed up and not presented in a logical order. The history is mixed with symptomatology etc.
- Elements from the discussion chapter have been moved to the case report section and references with discussions have been added to the case presentation, which is not allowed.
- The discussion chapter, on the other hand, has been completely altered. Almost nothing relevant has been added from the literature and half of the discussion chapter is related to the materials and methods, as well as limitations of the study.
- The conclusions were merely continued discussions. For example: "But in this case, it could not be applied due to national insurance issues and additional medication could not be administered due to side effects." What is the conclusion?
- The bibliography contains many irrelevant articles for the presented case or misinterpretations. For example: "34 - in the text it appears - Bifidobacterium and Ligilactobacillus are well-known probiotic bacteria{34}." However, the cited article is about bacterial flora in PEDIATRIC patients with bronchial asthma, while the case report is about Inflammatory Bowel Disease in ADULTS... etc.
Minor editing of English language required.
Reviewer 2 Report
Comments and Suggestions for Authors
I am very pleased with the major improvements in the new version of the manuscript. The Authors followed my suggestions/comments and now the manuscript appears clear and tidy and its importance is now well understood. Just a minor comment – Introduction – line 42 – please insert, as I wrote in my first review – reference Turner D, et al. 2021, about STRIDE-II.
Comments on the Quality of English LanguageGood. Just some minor editing is required.
Author Response
We greatly appreciate the time and effort you took to provide us with your valuable comments. Your feedback was instrumental in improving the quality of our manuscript, and we are grateful for your input. Additionally, we have included the reference that you requested, which has helped to strengthen our argument and support our conclusions. Thank you again for your contribution to our work.
Round 3
Reviewer 1 Report
Comments and Suggestions for Authors
The authors have made very few improvements compared to the previous version and with superficial changes that have not significantly increased the article's quality.
Comments on the Quality of English LanguageMinor editing of English language required.
Author Response
Thank you for your valuable comments.